# Impact Toughness of FRTP Composites Produced by 3D Printing

**DOI:** 10.3390/ma13245654

**Published:** 2020-12-11

**Authors:** Milan Vaško, Milan Sága, Jaroslav Majko, Alan Vaško, Marián Handrik

**Affiliations:** 1Department of Applied Mechanics, Faculty of Mechanical Engineering, University of Žilina, Univerzitná 8215/1, 010 26 Žilina, Slovakia; milan.vasko@fstroj.uniza.sk (M.V.); milan.saga@fstroj.uniza.sk (M.S.); marian.handrik@fstroj.uniza.sk (M.H.); 2Department of Materials Engineering, Faculty of Mechanical Engineering, University of Žilina, Univerzitná 8215/1, 010 26 Žilina, Slovakia; alan.vasko@fstroj.uniza.sk

**Keywords:** impact test, experimental measurement, additive manufacturing, FRTP composites, Kevlar fibre, carbon fibre, glass fibre, HSHT glass fibre

## Abstract

The additive manufacturing represents a new production method of composites reinforced with a continuous fibre. In recent times, the material produced by this new manufacturing method constituted a replacement for conventional materials—e.g., steel in many technical areas. As the research on FRTP composites is currently under way, the purpose of this article is to add information to the mosaic of studies in this research area. The scientific articles published until now have focused especially on mechanical testing, such as tensile and bending mechanical testing and their assessment. Therefore, the authors decided to carry out and assess the impact test of the FRTP composites produced by 3D printing because this area offers a large extent of research activities. We observed the influence of the reinforcement in the form of the micro-fibre carbon in the thermoplastic (Onyx) or a continuous reinforcement fibre in the lamina on the specimen’s behaviour during the impact load processes. The results of the experimental measurements show that the presence of a continuous fibre in the structure significantly affects the strength of the printed specimens; however, the design process of the printed object has to take into account the importance of selecting a suitable fibre type. The selection of a suitable strategy for arranging the fibre in the lamina and the direction of the impact load against the position of the fibre seem to be very important parameters.

## 1. Introduction

Industrial production has been accompanied by development and innovations of manufactured products and manufacturing technologies with the primary aim of overcoming the implemented limits, for the whole history. Due to this fact, innovations are an important aspect of competitiveness, which leads to the cycle of acquiring and implementing new knowledge in the practice. In general, the main goal is to be better than the others and, in this way, to achieve the competitive advantage against other market participants. In the technical segment, the development of new technologies and materials in the industry belongs to the important aspects [1].

Fibre reinforced thermoplastics (FRTP) are one of these materials. They gradually found application in many industries (e.g., in the automotive industry [2,3] but also especially in aviation and space technology [4,5,6]). They are composite materials consisting of a thermoplastic matrix reinforced with high-strength fibres (e.g., glass, carbon, Kevlar, etc.) The formed composite achieves better mechanical properties compared to individual materials. In the case of fibres, the main function is to reinforce the object in the form of a long or short fibre. Generally, long fibres result in better mechanical properties than short fibres. The matrix performs functions such as protection of the fibre from the environment, precise placement of the fibre in the structure, load transfer to the fibres, etc.

Their advantage is the ability to achieve the required material qualities specified by the industry, e.g., strength, toughness, resistance against corrosion, etc. with a low product weight [7,8,9]. In their application, the designer must take into account, for example, the limitations of shapes, the difficulty of predicting the behaviour of the material under the influence of physical phenomena, etc. However, the currently most important limitation for implementing the FRTP composites into practice is their complex production process, which presents a lot of problems [10,11,12,13].

Due to the problematic conventional method of production of FRTP composites, additive technologies originally focused only on plastic printing are coming to the foreground [14,15,16]. Only three main 3D printing methods allow the insertion of a fibre into the matrix from a technological point of view. One of them is the Fused Filament Fabrication (FFF) method [17].

The FFF method is based on the process of extruding a material, heating it and storing it in a predefined place [18,19,20]. The main parts of the printer are the feeder, the printer head, the nozzle and the work platform. The printer operation is based on the following steps. The feeder uses rollers to move the fibre of material into a nozzle, where it is heated to a predefined melting temperature. The printer head moves along a predefined path so that the molten fibre is placed in the desired location. During application, the molten material heats the previously deposited layers of material and solidifies due to cooling. At the same time, a bond is formed between the layers of material. When the storage of material in the entire layer is completed, the platform drops, and the cycle is repeated by printing the next layer until the entire object is printed. The method allows printing composites reinforced with short fibre [21,22,23].

Long fibre reinforced composites can be printed using the Continuous Fibre Fabrication (CFF) method [15]. This is an extended version of the FFF method, as the printer is equipped with two nozzles, one for laying a thermoplastic (nylon, ONYX) and the other for storing a long reinforcing fibre (carbon, glass, aramid, etc.) The principle of the method is shown in Figure 1.

Thanks to the controlled deposition of the fibre into the structure, significant accuracy of its deposition and volume fraction in the structure is achieved. These adjustments lead to achieving customisation and prototyping the designed objects for practice. The printed composite products can achieve mechanical properties comparable with the products made from metals, achieving high shape variability of the printed objects according to the requirements suitable for practical utilisation [24,25,26].

Since the CFF method was patented in 2015 and the additive production of the FRTP composites is still under way, the fibre’s influence on the structure behaviour under various loading conditions is currently investigated. The most frequently observed loading methods have been the tensile, compression and bending tests until today [27,28,29,30,31,32]. Due to the possible implementation of these materials in operation where damages caused by impacts with other objects may occur, it is also necessary to observe the behaviour of this material during impact test.

There are many unanswered questions in determining the effect of impact loading on 3D printed specimens according to a review study [33]. The study highlights the importance of predictive models (e.g., FEA) for the purpose of estimating the behaviour of printed composites and optimizing their mechanical properties. Numerical analyses show that in the case of impact tests, they are close to experimental measurements [34]. Implemented impact tests have shown that the thickness of the lamina, the volume fraction of the fibre and the build orientation have an effect on the amount of absorbed energy. In the case of nylon (not nylon reinforced with chopped carbon fibre—Onyx), the impact toughness depends on the lamina thickness. When the specimens were reinforced with fibre, the best values were obtained with the application of glass fibre. The considered fibre deposition was only isotropic and rectangular infill with 100% fill density [35]. Another study looked at the impact toughness of Onyx specimens by observing parameters such as lamina thickness, specimen orientation and solid fill (only rectangular, hexagonal and triangular). Compared to nylon, Onyx has been shown to be more brittle. This can be explained by the presence of chopped carbon fibre within Onyx. In addition, the test specimens were reinforced with continuous Kevlar fibre in an isotropic and concentric alignment. The presence of a continuous fibre led to an increase in the absorbed energy [36].

Therefore, the authors’ goal was to observe the influence of various parameters of printing (lamina thickness, fibre type, filament arrangement) on the impact toughness of the composite specimens made of the composite reinforced by a short or continuous type of the fibre. In the case of specimens made from nylon reinforced with carbon fibre, experimental measurements were performed on specimens with a solid infill with fill density of 100%. In this case, the influence of various parameters on the amount of absorbed energy was monitored (base orientation, number of walls, filament orientation). In addition, an analysis of the effect of these parameters on specimen weight, printing time, and printing cost was performed. The same experimental test was performed on specimens of nylon with chopped carbon fibre (Onyx) reinforced with continuous fibre (glass, HSHT glass, Kevlar, carbon). The main monitored parameters were the influence of the selected type of continuous fibre and the strategy of its deposition in the structure on the amount of absorbed energy. Subsequently, an analysis of the effect of these settings on sample weight, printing time and production costs was performed.

## 2. Impact Bending Test

The origin of the Charpy impact bending test was motivated by the need to complete the data about the basic mechanical properties determined by the tensile test with a criterion that would take into account the resistance of the material to brittle failure during the impact load. The test procedure is as follows: We place the test specimen of the determined shape and dimensions on a plate of the testing device, and the impact testing hammer breaks it by one stroke. The data read from the testing device (Charpy hammer), the dimensions and shape of the testing bar and fracture surface are the basis for the test assessment whose objective is to state the material resistance against the impact load, i.e., detecting the absorbed energy consumed for breaking the bar. The absorbed energy (J) required to break the test specimen is evaluated according to the currently valid norm. Notch toughness as a proportion of the energy consumed to break the sample and the cross-sectional area (J/m^2^, J/cm^2^) has been used in the past.

The impact is realised by the Charpy hammer with a gravitational force *F_G_* that is clamped in the height *H* before the test. After its release, it moves along a circular path, and its total potential energy changes into the kinetic one. It achieves its maximum in the bottom position, i.e., at the moment of the hammer’s impact on the testing bar. Part of the total potential energy is consumed for deformation and fracture of the testing bar, and the remaining part diverts the hammer trough the vertical position to the height *h*. The energy consumed for breaking the testing bar is given by the difference of the potential energies:(1)$$K=F_{G}\cdot \left(H-h\right)$$
where *F_G_* is the hammer´s gravitation force, *H* is the height of the hammer before the test, and *h* is the height of the hammer after the test. The absorbed energy *K* expresses the resistance of the material against the impact load. It is expected that the highest possible value of absorbed energy will be achieved. The magnitude of the absorbed energy is directly proportional to the deflection of the hammer from the lower vertical position to the height *h*. The absorbed energy is no physical quantity but an agreed comparing quantity. In spite of that, the impact test is important because the test results respond very sensitively on the fine difference in the structure that can be hardly observed by other mechanical tests. The standardised testing bar for the impact test is 55 mm long and has a square cross-section with sides of 10 mm. In the middle of the testing bar, there is a U-notch (or V-notch) for the metallic materials.

The tests of the composite materials have to fulfil the condition that the assessments and comparisons of the test results have to be realised on the testing bars of the same shape and dimensions. There is no general methodology for re-calculating the results achieved by one testing methodology to the values acquired by another testing methodology.

In the case of the impact bending tests, specimens of the composite material need not be notched. Because previous studies [35,36] were performed on notched specimens, the authors considered a similar type of specimen. The authors decided to perform the experiment on non-notched samples after careful consideration. The reason is the significant influence of the notch on the mechanical behaviour of the specimen under impact loading and the problematic deposition of reinforcement in the notch area. The fibre rings would copy the shape of the notch in the case of a notched sample. The strength of the material in the notch area will be reduced due to the change in the direction of the fibre and the fibres would be loaded by bending stress. The deposition of the fibre around the notch will show inhomogeneity (Figure 2).

The absorbed energy is significantly dependent on the test conditions, in particular, the test temperature. By reducing the test temperature in a certain temperature interval, the absorbed energy decreases from the maximum value of *K_max_* to the minimum value of *K_min_*. This interval is called the transition area. By lowering the operating temperature below the transition temperature, the plasticity drops sharply, and the type of failure changes from tough to brittle. It is important for technical practice that the construction material is loaded at temperatures higher than the transition temperature.

The Charpy impact bending test is carried out at the temperature of 23 ± 5 °C. If the test is realised at a different temperature than the ambient temperature, the bars have to be placed in a cooling or heating environment until the desired temperature in the whole cross-section of the testing bar is achieved. The testing bar has to be broken within 5 s after it is taken from this environment.

## 3. Preparation of the Impact Bending Tests

The Charpy impact bending test was carried out on a series of specimens with the geometric shape defined in Figure 3. In order to perform the measurement, we used the Charpy hammer PSV 300 with the nominal energy of 300 J. The specimens (55 mm × 10 mm × 10 mm) were printed by the 3D printer MarkForged Mark Two whose operation is based on the principles of the FFF and CFF technologies. Each test series for measurement consisted of five specimens.

Specified printer can print the materials listed in Table 1. Onyx is the trade name of a material that represents nylon in the form of a fibre filled with chopped carbon fibre. In the case of Chapter 4, the specimens were made only from Onyx by the FFF method. Chapter 5 deals with long fibre reinforced specimens (glass, carbon, Kevlar, HSHT glass). The specimen matrix consisted of Onyx. Printing of long-fibre reinforced specimens is only possible by the CFF method.

Various changes were made to the print settings before the actual printing in order to monitor the influence of the print parameters on the behaviour of the specimens under the impact load (Figure 4 and Table 2).

The selected material infill of material pattern in each lamina was solid fill.

The primary aim was to point out the influence of various printing parameters, e.g., the orientation of layers, material deposition, lamina thickness, the orientation of the specimen against the Charpy hammer on the impact toughness. The orientation of the printed laminas against the balance weight of the testing device is shown in Figure 5, Figure 6 and Figure 7.

## 4. Impact Bending Tests—Specimens Reinforced by Short Carbon Fibre in Thermoplastic

The measured values of the absorbed energy from the impact bending test of the specimens reinforced with short carbon fibres are shown in Table 3. The parameters—weight, volume, printing time and costs were acquired from the supporting commercial software provided by the manufacturer of the printer. Due to detecting the influence of various parameters, each series was different. Selected parameters (such as printing time and weight) were experimentally verified, and negligible differences were observed.

The comparison of the average value of the absorbed energy by individual specimens from the corresponding series shows a significant influence of all parameters. The specimen printed to the height with the base XZ absorbed the lowest volume of energy. On the contrary, the specimens with material placed in lamina under the angle of 45/−45 with the XY base achieve the best values of energy, i.e., those printed horizontally. The 45/−45-degree orientation has a higher impact resistance than the 0/90-degree orientation because the individual layers are better bonded with respect to the direction of the load. It means that all layers counter the crack propagation, while in a 0/90-degree orientation, only 50% of layers efficiently counter crack propagation. When we compare the specimens with different lamina thicknesses, the energy absorption depends on the orientation of the laminas against the load. In the case of the specimens with the lamina thickness of 0.1 mm, better results are achieved for the loads oriented perpendicularly to the layers. If the lamina thickness is 0.2 mm, the specimen absorbs more energy if it is oriented in such a way the direction of placing the layer is in compliance with direction of the impact [35,38,39,40]. The difference between the series No. 6 and No. 7 shows that the number of rings along the circumference of the specimen reduced the specimen toughness.

Since we observed considerable differences of the time volume that was necessary for printing of various types of specimen during the analysis, in addition, we decided to observe the influence of several known printing parameters. The main goal was to determine the most suitable arrangement of the material in the specimen or its suitable orientation regarding the impact direction from the point of view of not only the energy but also other criteria.

### 4.1. Inluence of Printing Parameters on Weight of Specimen

The specimen weight was the first monitored parameter in the complex analysis. In the case of this type of diagrams, the individual specimen types are defined in the legend as follows: serial number; material; lamina thickness; base plane; filament orientation. From the point of view of the weight, the differences between the specimens are negligible—the differences are at the level of 3%. When we compare the ratio of the energy absorbed by the specimen and its weight, the specimens from the series No. 7 show the best results (7.5 J). The worst results were achieved for the specimen printed vertically (1.5 J).

All observed printing parameters affect the weight of the specimen (Figure 8). We can sort them according to their importance as follows:Filament orientation;Thickness of lamina;Base plane;Number of walls.

The greatest influence is shown by the different deposition of material in the layer. In general, the specimens with the filament deposited under the angles 0/90 are lighter. That is valid for both the specimens printed horizontally and vertically. This is due to the set G code from the manufacturer. The storage paths and the amount of extruded material are defined in order to fill the entire surface of the lamina with material. The same conclusions are also for the volume of the consumed material.

### 4.2. Influence of Printing Parameters on Printing Time of Specimen

The time is another significant parameter for the 3D printing process. The duration of the printing process during creating these specimens could be doubled in dependence on the adjusted printing parameters. The printing procedure of the specimens from the series 6 (38 min), 7 and 8 (35 min) took the shortest time.

All observed print parameters have an effect on the sample weight (Figure 9). According to their importance, we can rank them as follows:Thickness of lamina;Base plane;Filament orientation;Number of walls.

The lamina thickness is the main parameter, which significantly affects the duration of the printing process. The change of the lamina thickness from 0.2 to 0.1 mm causes the printing duration to be almost twofold (35 versus 66 min).

The last considered parameter was printing costs. The differences between the costs necessary for the printing of individual specimen type vary maximally by 3%. Lower costs are for samples with an Onyx configuration at an angle of 0/90. However, with regard to the absorbed energy of samples with different layering, the samples from series No. 7 are remarkable.

## 5. Impact Bending Tests—Specimens with Continuous Fibre Deposited in Thermoplastic Structure

After realising the analysis on the specimens reinforced with the short carbon fibre, we carried out the same tests on the specimens reinforced by the continuous fibre (Figure 10) and observed the same parameters—weight, volume, time, printing costs and the amount of absorbed energy. The specimens were reinforced by the carbon, Kevlar, glass and HSHT glass fibre that can be arranged in a 45-degree orientation, longitudinally, in concentric rings or in a mixed way in the structure (Figure 11). We selected the following strategies of arranging the fibre in the specimen (Table 4).

The selected matrix material was nylon reinforced with chopped carbon fibre (Onyx trademark), whose deposition strategy used in the case of continuous fibre reinforced thermoplastic (CFRTP) composites is shown in Figure 4d,e. The number of the reinforced layers with the continuous fibre varied as it is dependent on the lamina thickness. The lamina thickness is a parameter adjusted by the printer manufacturer for each type of the reinforcing fibre. In the case of the glass, HSHT glass and Kevlar fibre, the thickness of one layer can be only 0.1 mm, i.e., the specimen was divided to 100 laminas. In order to assess the effect of the presence of fibre on the impact toughness, samples reinforced with Kevlar, glass or HSHT glass fibres were divided into two groups according to the number of reinforced layers (Table 4).

Two configurations were studied: (I) a “sandwiched” configuration with 8 reinforced layers equally distributed in the vertical direction (on a total of 100 layers—Figure 10a); (II) a “plain” configuration with only reinforced layers (92 layers in total—Figure 10b). Both configurations have a top and bottom part made of 4 layers of nylon matrix.

As the thickness of one lamina is adjusted for the value of 0.125 mm when the carbon fibre is used, the slicing software divided the specimen reinforced by a carbon fibre into 80 laminas. As the individual fibre types were to be compared, we kept the equivalent number of then reinforced laminas. Due to this requirement, the continuous carbon fibre was deposited in 6 or 72 layers.

The loading method of the specimens of this type was similar as in the case of the specimens reinforced by the chopped cut fibre (see Figure 5 and Figure 6) with the goal to observe if the printed composite behaves 45-degree oriented and how various parameters of the fibre affect this behaviour (deposition, orientation, etc.)

The results of measurement and detected data from slicing software are shown in Table 5. The series of specimens 11 and 12 are taken over from the measurement analysed in the previous section to compare the reinforced and non-reinforced specimens.

The results show the difference of the impact toughness between individual specimens is considerable. In general, the types of used fibres can be ranked as follows, according to their effect on the amount of the absorbed energy:HSHT glass fibre;glass fibre;Kevlar fibre;carbon fibre.

The difference between the toughness of the specimens reinforced by the HSHT glass fibre and the Kevlar fibre is twofold in favour of the first fibre type. In the case of the difference between the HSHT glass fibre and the carbon fibre, this difference is even more considerable—even fourfold.

As it was shown, the results were significantly affected just by the arrangement of the fibre in the structure, we additionally selected other strategies of its arrangement. First of all, they are the concentric rings of the HSHT fibre in the structure. When we compare the series 7, 8, 9 and 10, we can see that the concentric fibre rings in the structure enable absorbing a higher amount of energy than the longitudinally deposited fibres. Then, we considered the isotropic deposition of the fibre under the angles 45/−45 and 30/−60/60/45/−45. However, the achieved values of absorbed energy show that this is not the most suitable fibre storage configuration for such a method of loading.

The orientation of the layers with respect to the force acting on the specimen must also be taken into account in addition to the selection of the appropriate type of fibre and its arrangement in the lamina. The effect of this parameter is visible in the case of a significant difference in the absorbed energy between samples No. 19 and 20. In the case of the longitudinal deposition of the fibre with one concentric ring and high FVF of all fibre types, better values of toughness are achieved for the perpendicular loads of the specimen on the direction of depositing the laminas. The specimens with a low volume of the HSHT glass fibre and carbon fibre in the structure that achieve a higher strength at the direct load are an exception.

The largest amount of the absorbed energy was in the case of the specimen reinforced with the HSHT fibre with a concentric arrangement at the level of 48 J. Compared with previous types of arrangement, this strategy developed a higher absorption of energy when the lamina was loaded directly. The values of the absorbed energy show that compared with the conventionally produced composites for which just the fibre arrangement of 45/−45 is the most suitable choice [41], the concentric rings for the specimen produced by the additive technologies are the most appropriate strategy of the fibre arrangement. The highest achieved values of the impact strength are comparable with some types of steels or aluminium alloys [24].

Figure 12 depicts the broken specimens reinforced with the individual types of fibre.

The assessment of suitability of the individual fibre types for various applications in practice can be considerably limited by their different physical properties; therefore, each type of fibre and its arrangement in the structure was assessed from the point of view of some important parameter (such as costs, fibre volume fracture (FVF), time, etc.)

### 5.1. Influence of Printing Parameters on Weight

Compared with the specimens reinforced only with a chopped carbon fibre, in the case of the CFRPT composite, we can see a principal influence of the selected printing parameters in the specimen weight.

In general, the specimen weight mostly increases when the fibres are present in the structure. The Kevlar fibre is an exception—it has a comparable density as the material that creates the matrix (e.g., nylon reinforced with a short carbon fibre). Due to this fact, we can explain the phenomenon that the increased number of the reinforced layers reduced the specimen weight when this type of fibre was used. Specimens reinforced with HSHT glass fibre deposited in concentric rings reached the highest weight.

The decrease in the weight of the specimens is less influenced by the chosen fibre deposition strategy, while the decrease in absorbed energy is not so significant (Table 6). Concentric rings are among the most suitable fibre arrangement strategies (or even longitudinal deposition). A decrease of weight to the level of the longitudinally deposited fibre in laminas was observed in the case of the 45-degree orientation fibre arrangement, but according to the results (except for HSHT 45/−45 perpendicular), the toughness of the specimen was considerably affected. Due to this fact, the 45/−45 arrangement of the fibre in the structure does not seem as an optimal strategy.

From the point of view of selecting a suitable fibre with regard to the absorbed energy (Figure 13), the order of the fibres is as follows:HSHT glass fibre;Glass fibre;Kevlar fibre;Carbon fibre.

The best ratio is achieved with HSHT glass fibre. A little worse ratio can be achieved with the glass fibre. However, a lower toughness of the specimens was achieved with the glass fibre. In the case where the weight was the most important parameter of the printed object, we could also think about using the Kevlar fibre.

### 5.2. Influence of Printing Parameters on Fibre Volume Fracture

Except for the weight, another observed parameter is the volume share of the fibre (fibre volume fraction) in the structure. Based on observing the relation between FVF and energy, we can say that in general the specimen toughness is increased with the enlarged fraction of the fibre in the structure.

The Kevlar and carbon fibres with the volume share 4.6% and 5.1%, respectively, are an exception. The energy necessary for perforating the specimen was smaller than in the case where it was not reinforced with the continuous fibre. From the point of view of the fibre arrangement in the structure, the best results were achieved when we used the strategy of the concentric rings. Compared with the longitudinal fibre arrangement, there is a lower consumption of the fibre because Onyx is placed in the middle of the specimen (Figure 11b). From this point of view, a suitable alternative can be the arrangement 45/−45 in the case of the glass fibre at the perpendicular load. Adjusting the fibre arrangement in the structure has the greatest effect on FVF size and fibre consumption (Table 7).

The lowest fibre consumption was achieved using a 45-degree orientation fibre deposition strategy. However, sufficient values of absorbed energy were achieved only in the case of the perpendicular type of loading. The concentric ring reinforcement seems to be the most suitable for this reason.

### 5.3. Influence of Printing Parameters on Printing Time

The production time is an important parameter for manufacturing. In this case, Table 3 shows the fact that the application of the fibre in the structure led to a considerable growth of the printing time. The deposition of the Kevlar fibre to 8 layers (to total number 100) led to increasing the printing time from 35 to 90 min.

The carbon fibre requires the shortest printing time of all fibre types, but in this case, we have to take into account the pre-adjusted thickness of one lamina by the printer producer that could have a significant influence on it, according to the values for Onyx from Table 3. The specimens reinforced with the HSHT glass fibre require the longest printing time. Especially, the high volume of the HSHT glass fibre in the structure leads to a significant prolongation of the printing time (the printing time of approximately 160 min). Compared with the non-reinforced specimens, the printing time is excessively prolonged (about five times).

From the point of view of the fibre arrangement—the specimens with the HSHT fibre arrangement 45/−45 and 60/30/30/60/45/−45 take the longest printing time—the printing time of one specimen lasts more than 200 min. The probable reason is the different composition of the lamina structure. In the case of the arrangement 45/−45, gaps develop between the specimen walls and the fibre, and they have to be filled with Onyx (Figure 11c). Other strategies for HSHT fibre result in a 15% decrease in printing time. When monitoring the effect of the deposition strategy in the case of carbon fibre, the decrease was at the level of 13.6%. This is a relatively comparable value to HSHT glass fibre. The concentric ring strategy is therefore the best choice given the amount of the absorbed energy (Figure 14). To select a suitable fibre type, glass fibre is offered as a good alternative to HSHT glass fibre (Table 8).

### 5.4. Influence of Printing Parameters on Production Cost

The manufacturing costs are one of the most important parameters. The high manufacturing costs are the reason why the 3D print as a new technology is implemented in practice only step by step. Therefore, the authors carried out a brief analysis of the relation between the production costs and the investigated mechanical properties of individual types of the testing specimen at the end of this article.

From the point of view of the costs, the specimen without the continuous fibre in the structure is the cheapest variant. The disadvantage of this variant is the too low impact strength of the specimens. The highest manufacturing costs are connected with adding the continuous carbon fibre to the structure. in spite that the specimens reinforced with the carbon fibre achieved the worst results out of all reinforced specimens (Figure 15).

There is a significant difference between the individual types of fibres in terms of cost. The lowest costs were achieved when applying glass fibre. HSHT glass fibre is also a suitable alternative with regard to the amount of absorbed energy. The carbon fibre is not suitable for this type of loading due to too high costs and low amount of absorbed energy.

The fibre storage strategy generally has less impact on cost than the choice of fibre type. The lowest production costs were in the 45-degree orientation arrangement of the fibre in the structure. However, when monitoring the effect of stratification parameters on the amount of absorbed energy, the application of concentric rings is a more suitable alternative (Table 9).

Furthermore, a comparison of the increase in cost and absorbed energy of selected fibre types was performed in comparison with specimens reinforced with short fibre (Table 10). The table shows that the application of carbon fibre is not suitable.

The cases of the specimens reinforced with other types of fibres show the costs are more or less comparable. The comparison of the HSHT and Kevlar fibres results show a twofold difference of the absorbed energy to disadvantage of the Kevlar fibre. It means the usage of the Kevlar fibre is less effective. The application of the glass fibre especially in the case of the perpendicular load is a suitable alternative (the manufacturing costs are also lower). Summarising, the best results are achieved by placing the HSHT glass fibre into the structure also from the viewpoint of the costs.

In the case of several arrangement types, the ratio of the increase of the energy absorbed overcame the ratio of cost increase for producing these specimens. From this viewpoint, all three strategies of the fibre arrangement (45/−45, concentric rings, longitudinal) appear to be very effective for the perpendicular load. In the case of the direct load, the highest value of the absorbed energy was provided by the strategy of the concentric rings along the specimen circumference (48 J).

When considering all parameters (weight, cost, time, consumption), the specimen reinforced with HSHT glass fibre embedded in the structure in the form of concentric rings appears to be the best. In some cases, better values were obtained with other types of fibres. HSHT glass fibre achieved relatively comparable results even in such cases. The specimens reached the highest values of absorbed energy when this fibre was applied in the structure.

## 6. Conclusions

The objective of this article was to present the results of the impact tests realised on the testing specimens manufactured by the 3D print from thermoplastic reinforced with the carbon chopped fibre (Onyx) and Onyx reinforced with the continuous fibre. Based on the impact bending tests realised on Onyx, we can say the specimens reinforced with the chopped carbon fibres achieve the values of the impact strength of very low values—maximally 7.5 J.

The thickness of one lamina of the specimens reinforced with the carbon chopped fibre seems to be a significant factor. However, the suitable selection of the lamina thickness depends on the load direction. It means the properties of this material are dependent of the direction. The lowest strength values are achieved for loading the testing specimen in the direction of printing.

The comparisons of the results according to the direction of printing show that the arrangement of the matrix material in lamina under the angle of 45/−45 provides better results compared with the 0/90 arrangement. The printing time is also a very important parameter. In dependence on the printing direction, the duration of the printing process can be considerably prolonged. Due to this reason, the specimens with the lamina thickness of 0.2 (series 6 and 7) can be considered the best choice.

In the case of Onyx reinforced with the continuous fibre, we can say that with an increasing proportion of the fibre, the absorption of energy during the impact test significantly grows. The highest toughness is achieved for the HSHT glass fibre reinforcement. In certain cases and when we consider a low weight, we can consider the usage of the glass or Kevlar fibre too. The composite structures reinforced with the carbon fibre seem to be unsuitable if we assume a similar type of loading.

The realised test shows that regarding the costs and printing time the concentric rings are the most suitable deposition method from the point of view of arranging the fibre. The longitudinal depositions of the fibre achieved also good results. In the 45/−45 arrangement we can recognise dependence between the specimen orientation and the corresponding direction of loading. Based on a thorough analysis of the test results, we observed a significant influence of the specimen´s orientation against the loading direction on the amount of the absorbed energy. Regarding to this fact we can assume that the printed specimens behave in an anisotropic manner during impact loading.

When we choose a suitable type of the reinforcing fibre, other criteria have also a significant influence. The selection of the fibre considerably affects various parameters—the specimen weight, printing time and the manufacturing costs. Although the HSHT glass fibre remarkably increases the toughness resistance of the specimen, its usage considerably increases the printing time and weight of the specimen for all types of depositing the fibre. On the other hand, lower costs are its advantage.

In the case of the composites manufactured by the 3D print, the most suitable method is to use the reinforcement created by the continuous fibres. The best results are achieved with the HSHT fibre when the values of the absorbed energy are comparable with the selected steel structures. That proves the suitability to use these reinforced composite structures in those areas where the steels have been dominant until today. The low weight is also an undisputable advantage of the composites compared with the steels.

## Figures and Tables

**Figure 1 materials-13-05654-f001:**
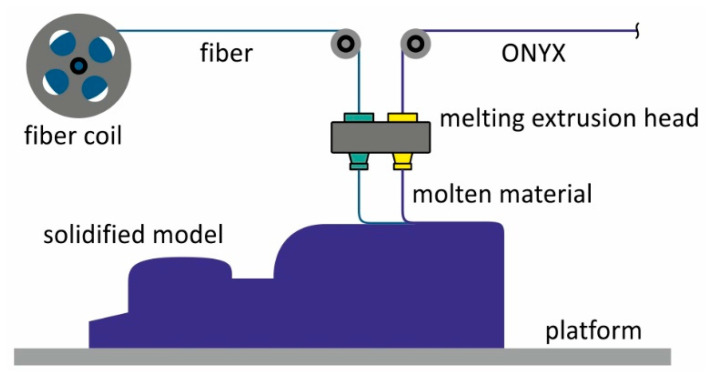
The principle of the Continuous Fibre Fabrication (CFF) method.

**Figure 2 materials-13-05654-f002:**
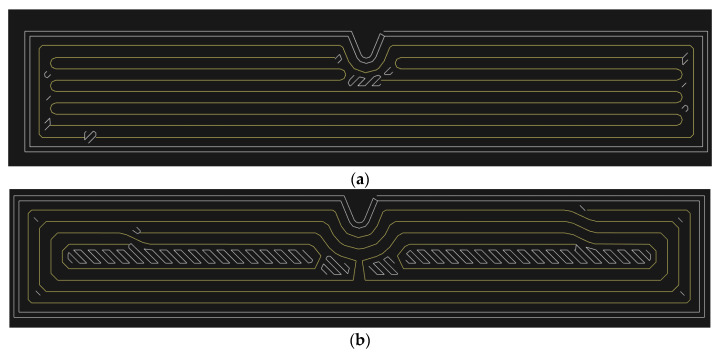
Notched specimen and fibre deposition in the notch area: (**a**) specimen reinforced with longitudinally deposited fibre and one concentric ring; (**b**) specimen reinforced with four concentric fibre rings.

**Figure 3 materials-13-05654-f003:**
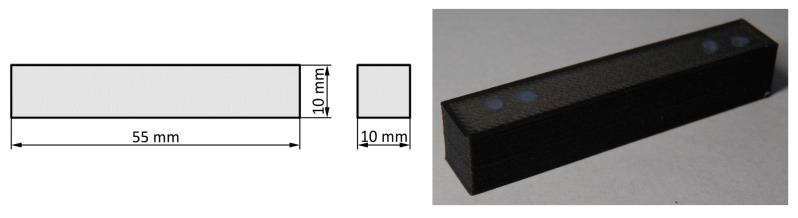
Shape of the specimen.

**Figure 4 materials-13-05654-f004:**
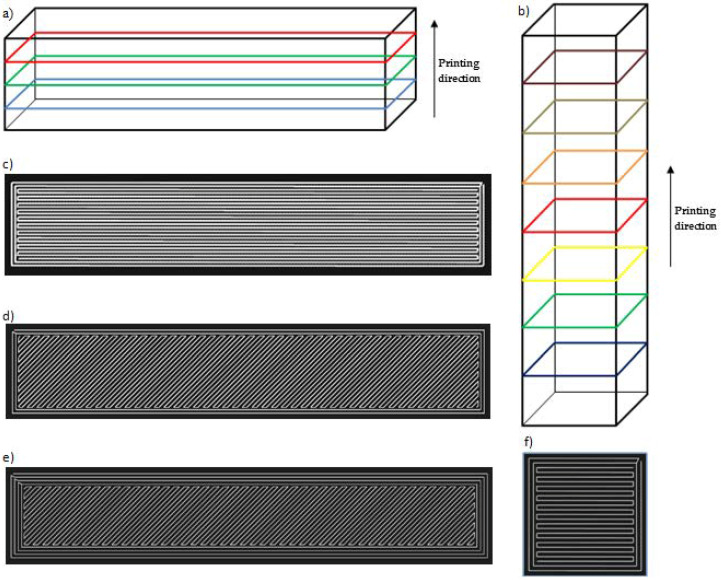
The printing direction and deposition of material: (**a**) the specimen with the base on the XY plane; (**b**) printing the specimen with the base on the XZ plane; (**c**) material deposition in the lamina under the angle 0/90 with two rings; (**d**) material deposition in the lamina under the angle 45/−45 with two rings; (**e**) specimen with four rings; (**f**) material deposition in the lamina under the angle 0/90 with two rings.

**Figure 5 materials-13-05654-f005:**
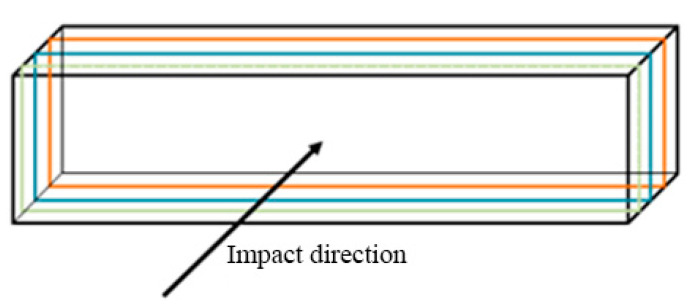
Loading direction—direct.

**Figure 6 materials-13-05654-f006:**
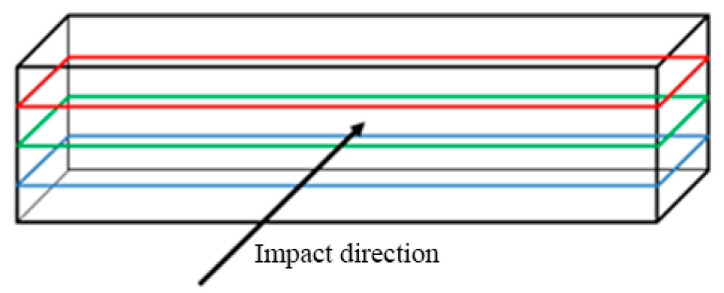
Loading direction—perpendicular.

**Figure 7 materials-13-05654-f007:**
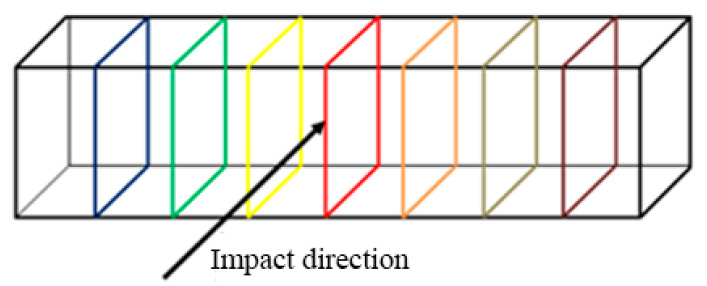
Loading direction—perpendicular.

**Figure 8 materials-13-05654-f008:**
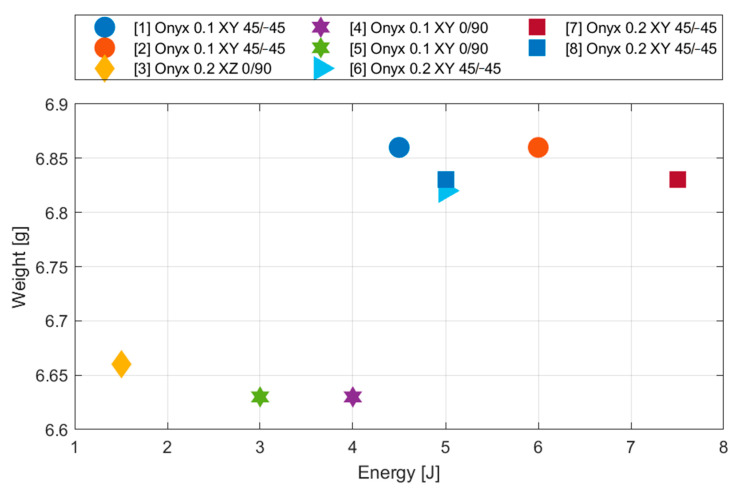
Influence of layering on specimen weight (comparison of absorbed energy for the purpose of appropriate specimen type selection).

**Figure 9 materials-13-05654-f009:**
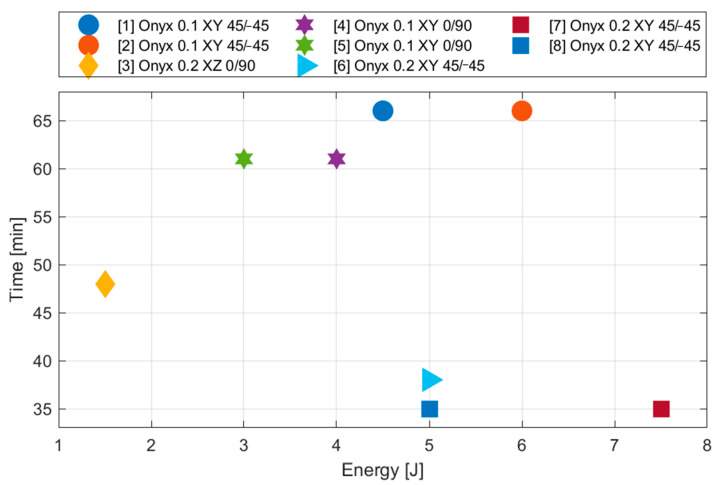
Influence of layering on printing time (comparison of absorbed energy for the purpose of appropriate specimen type selection).

**Figure 10 materials-13-05654-f010:**
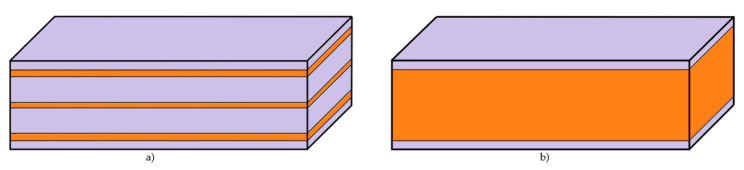
Configuration of reinforced layers in the structure of continuous fibre reinforced thermoplastic (CFRTP) composite (orange—fibre, purple—matrix): (**a**) Specimen reinforced with continuous fibres in 8 layers; (**b**) the specimen reinforced with continuous fibres in 92 layers.

**Figure 11 materials-13-05654-f011:**
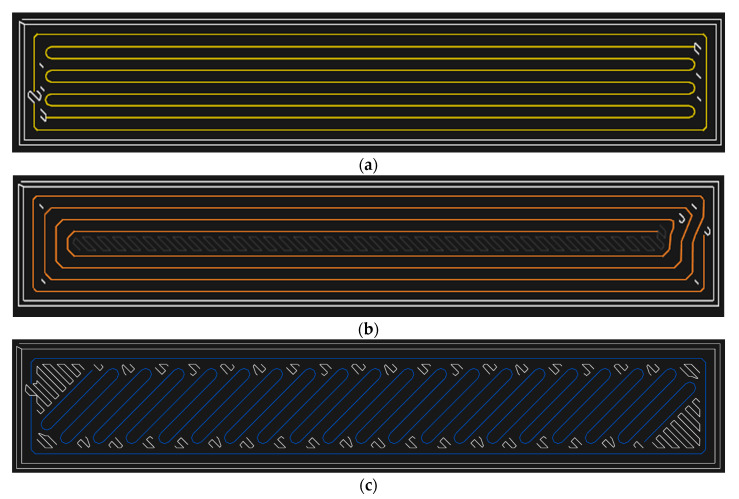
The method of arranging the fibre in the structure: (**a**) The longitudinally arranged fibre with one concentric ring; (**b**) the concentric rings; (**c**) isotropically arranged fibre with a defined direction of the fibre, e.g., 45°, 30°, 60° or combinations of angles.

**Figure 12 materials-13-05654-f012:**
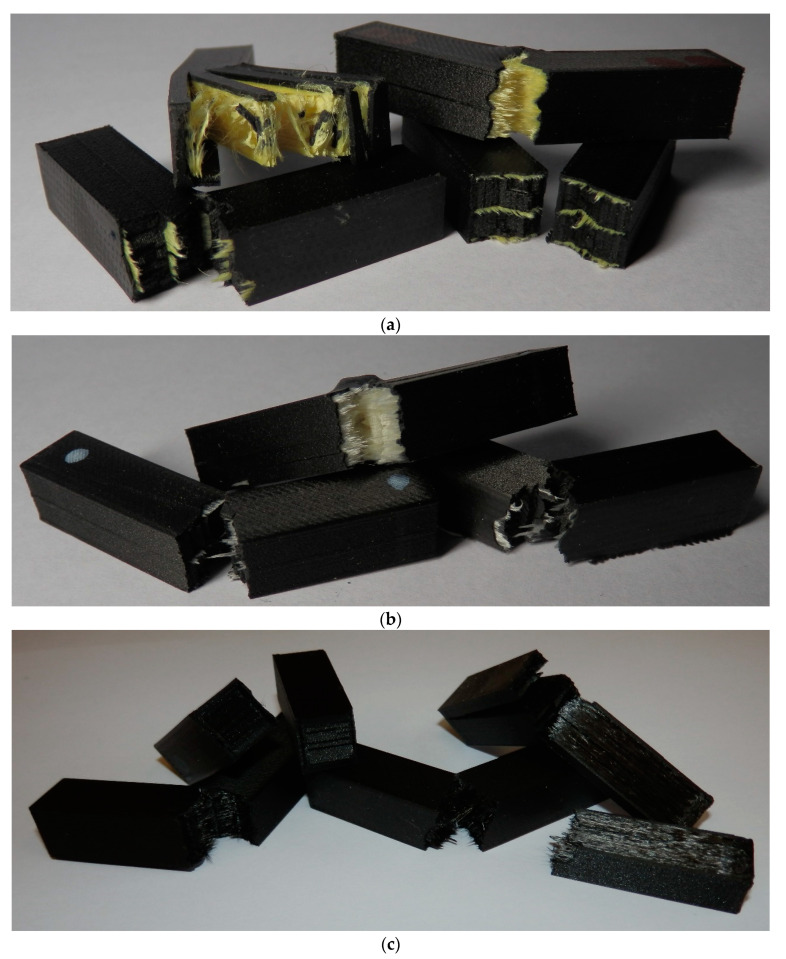
Specimens after the impact test: (**a**) Reinforced with the Kevlar fibre; (**b**) reinforced with the glass fibre; (**c**) reinforced with the carbon fibre; (**d**) reinforced with the HSHT glass fibre.

**Figure 13 materials-13-05654-f013:**
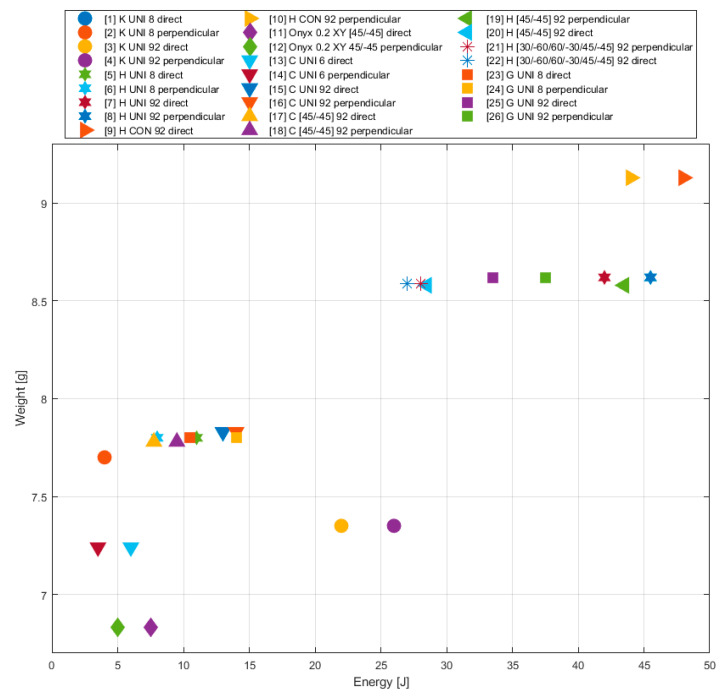
Influence of different printing parameters (fibre type, deposition strategy) on the weight of the specimen (comparison of absorbed energy for the purpose of appropriate specimen type selection).

**Figure 14 materials-13-05654-f014:**
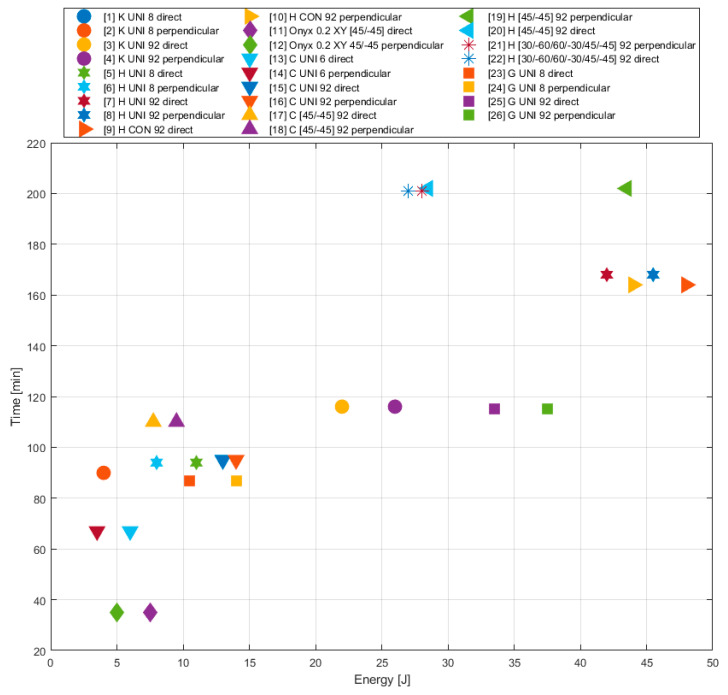
Influence of different printing parameters (fibre type, deposition strategy) on the printing time of specimen (comparison of absorbed energy for the purpose of appropriate specimen type selection).

**Figure 15 materials-13-05654-f015:**
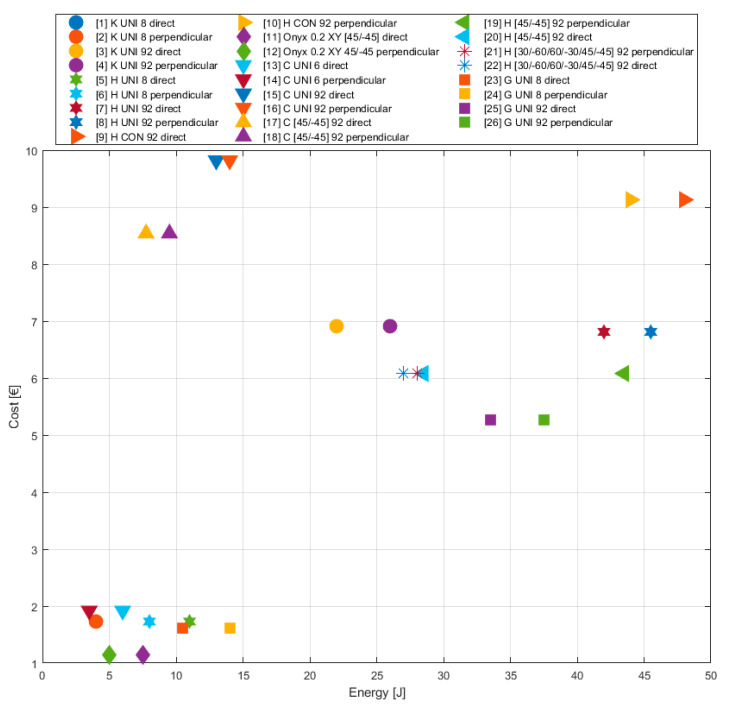
Influence of different printing parameters (fibre type, deposition strategy) on the production cost of specimen (comparison of absorbed energy for the purpose of appropriate specimen type selection).

**Table 1 materials-13-05654-t001:** Mechanical properties of the fibres [37].

	Nylon	Nylon Reinforced with Chopped Carbon Fibre (Onyx)	Glass Fibre	Kevlar Fibre	Carbon Fibre	HSHT Glass Fibre
Tensile Modulus (GPa)	1.7	1.4	21	27	60	21
Tensile Strength (MPa)	36	30	590	610	800	600
Tensile Strain at Break (%)	150	58	3.8	2.7	1.5	3.9
Flexural Strength (GPa)	50	81	200	240	540	420
Flexural Modulus (MPa)	1.4	3.6	22	26	51	21
Flexural Strain at Break (%)	-	-	1.1	21.	1.2	2.2
Printing method	FFF	FFF	CFF	CFF	CFF	CFF

**Table 2 materials-13-05654-t002:** Changed print parameters.

Parameter	Value
Layer thickness of matrix	0.1 or 0.2 mm
Base plane of specimen	XY (Figure 4a) or XZ (Figure 4b)
Matrix filament orientation in lamina	0°/90° (Figure 4c) or 45°/−45° (Figure 4d,e)
Number of walls	2 or 4
Number of rings	2 (Figure 4c,d) or 4 (Figure 4e)
Fill density	100%
Loading direction	Perpendicular or direct
Total number of laminas	50 or 100

**Table 3 materials-13-05654-t003:** Achieved average values of the absorbed energy by the specimen from each series and printing parameters defined by authors.

Serial Number	1	2	3	4	5	6	7	8
Thickness of lamina (μm)	100	100	200	100	100	200	200	200
Loading direction	Direct	Perpen.	Perpen.	Direct	Perpen.	Direct	Direct	Perpen.
No. of walls	2	2	2	2	2	4	2	2
Filament orientation	45/−45	45/−45	0/90	0/90	0/90	45/−45	45/−45	45/−45
Base plane	XY	XY	XZ	XY	XY	XY	XY	XY
Weight (g)	6.86	6.86	6.66	6.63	6.63	6.82	6.83	6.83
Volume (cm^3^)	5.81	5.81	5.65	5.62	5.62	5.78	5.78	5.78
Time (min)	66	66	48	61	61	38	35	35
Cost (€)	1.16	1.16	1.13	1.13	1.13	1.16	1.16	1.16
Absorbed energy (J) {(kJ/m^2^)}	4.5 {45}	6 {60}	1.5 {15}	4 {40}	3 {30}	5 {50}	7.5 {75}	5 {50}
Standard deviation (J)	0.55	0.22	0.22	0.55	0.27	0.7	0.89	1.14

**Table 4 materials-13-05654-t004:** Setting print parameters.

Parameter	Value
Fibre type	Kevlar, Carbon, Glass, HSHT Glass
Number of reinforced layers	Kevlar, Glass, HSHT Glass: 8 (Figure 10a) and 92 (Figure 10b) Carbon: 6 (Figure 10a) and 72 (Figure 10b)
Fibre deposition	A 45-degree orientation (Figure 11c), longitudinal with one concentric ring (Figure 11a), four concentric rings (Figure 11b)
Lamina thickness	Kevlar, Glass, HSHT Glass: 0.1 mm; Carbon: 0.125 mm
Loading direction	Direct (Figure 5) or Perpendicular (Figure 6)

**Table 5 materials-13-05654-t005:** Results of the impact bending test realised on the specimens reinforced by continuous fibres.

Ser.	Fibre Type ^1^	Deposition of Continuous Fibre ^2^	No. of Layer	Loading Direction	Weight (g)	FVF (%)	Time (min)	Cost (€)	Energy (J) {(kJ/m^2^)}	Standard Deviation (J)
1	K	L	8	Direct	7.7	4.6	90	1.73	4 {40}	0.42
2	K	L	8	Perpend.	7.7	4.6	90	1.73	4 {40}	0.22
3	K	L	Full	Direct	7.35	64.8	116	6.91	22 {220}	0.35
4	K	L	Full	Perpend.	7.35	64.8	116	6.91	26 {260}	1.48
5	H	L	8	Direct	7.8	4.46	94	1.73	11 {110}	0.57
6	H	L	8	Perpend.	7.8	4.46	94	1.73	8 {80}	0.97
7	H	L	Full	Direct	8.62	64.5	168	6.81	42 {420}	1.09
8	H	L	Full	Perpend.	8.62	64.5	168	6.81	45.5 {455}	0.84
9	H	C	Full	Direct	9.13	54.8	164	6.51	48 {480}	0.45
10	H	C	Full	Perpend.	9.13	54.8	164	6.51	44 {440}	0.55
11	---	None	None	Direct	6.83	0	35	1.15	7.5 {75}	0.89
12	---	None	None	Perpend.	6.83	0	35	1.15	5 {50}	1.13
13	C	L	6	Direct	7.24	5.1	67	1.92	6 {60}	0
14	C	L	6	Perpend.	7.24	5.1	67	1.92	3.5 {35}	0.45
15	C	L	Full	Direct	7.83	62.9	95	9.82	13 {130}	0.61
16	C	L	Full	Perpend.	7.83	62.9	95	9.82	14 {140}	0.91
17	C	45/−45	Full	Direct	7.78	52.9	110	8.54	7.7 {77}	0.27
18	C	45/−45	Full	Perpend.	7.78	52.9	110	8.54	9.5 {95}	0.42
19	H	45/−45	Full	Perpend.	8.58	54.4	202	6.08	43.5 {435}	0.42
20	H	45/−45	Full	Direct	8.58	54.4	202	6.08	28.5 {285}	1.13
21	H	30/−60/60/ −30/45/−45	Full	Perpend.	8.59	54.3	201	6.08	28 {280}	0.27
22	H	30/−60/60/ −30/45/−45	Full	Direct	8.59	54.3	201	6.08	27 {270}	0.71
23	G	L	8	Direct	7.8	4.46	87	1.61	14 {140}	0.65
24	G	L	8	Perpend.	7.8	4.46	87	1.61	10.5 {105}	0.45
25	G	L	Full	Direct	8.62	64.5	115	5.26	33.5 {335}	0.65
26	G	L	Full	Perpend.	8.62	64.5	115	5.26	37.5 {375}	0.57

^1^ Legend: K = Kevlar fibre, H = HSHT glass fibre, C = Carbon fibre, G = Glass fibre, --- = Onyx without any type of continuous fibre; ^2^ Legend: L = Longitudinal, C = Concentric.

**Table 6 materials-13-05654-t006:** Influence of fibre deposition and fibre type on specimen weight reduction.

Fibre Type	Weight Reduction	Deposition Strategy	Weight Reduction	Absorbed Energy Reduction
HSHT glass	-	4 concentric rings	-	-
Glass	0%	Longitudinal with one concentric ring	−5.6%	−5.2%
Carbon	−9%			
Kevlar	−15%	45-degree orientation	−6%	−9.38%

**Table 7 materials-13-05654-t007:** Influence of selected fibre deposition strategy on fibre consumption and fibre volume fraction.

Fibre Deposition Strategy	Fibre Consumption(cm^3^)	Fibre VolumeFraction (%)
45-degree orientation (Figure 11c)	3.31	54.4
Four concentric rings (Figure 11b)	3.55	54.8
Longitudinal with one concentric ring (Figure 11a)	3.83	64.5

**Table 8 materials-13-05654-t008:** Influence of printing parameters on printing time.

Fibre Type	Printing Time Reduction	Absorbed Energy Reduction	Deposition Strategy	Printing Time Reduction
HSHT glass	-	-	45-degree orientation	-
Kevlar	−31%	−42.9%	Longitudinal with one concentric ring	−16.8%
Glass	−31.5%	−17.6%		
Carbon	−43%	−69.2%	4 concentric rings	−18.8%

**Table 9 materials-13-05654-t009:** Influence of printing parameters on production cost.

Fibre Type	Cost Reduction	Deposition Strategy	Cost Reduction
Carbon	-	Longitudinal with one concentric ring	-
Kevlar	−29.6%		
HSHT glass	−30.6%	4 concentric rings	−4.4%
Glass	−46.4%	45-degree orientation	−10.7%

**Table 10 materials-13-05654-t010:** Ratio of the cost increase and growth of the absorbed energy for individual specimen types.

Selected Series	Fibre	Fibre Arrangement	Cost Growth Ratio for Selected Reinforced Specimen	Absorbed Energy Growth Ratio for Selected Reinforced Specimen
3	Kevlar	Direct	5.99	2.93
4	Kevlar	Perpendicular	5.99	5.2
7	HSHT	Direct	5.92	5.6
8	HSHT	Perpendicular	5.92	9.1
9	HSHT	Direct	5.66	6.4
10	HSHT	Perpendicular	5.66	8.8
15	Carbon	Direct	8.49	1.73
16	Carbon	Perpendicular	8.49	2.8
19	HSHT	Perpendicular	5.27	8.7
20	HSHT	Direct	5.27	3.8
25	Glass	Direct	4.57	4.46
26	Glass	Perpendicular	4.57	7.5

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
