# Peer review of "Impact Toughness of FRTP Composites Produced by 3D Printing"

_materials, 2020, doi:10.3390/ma13245654_

Round 1
Reviewer 1 Report
Brief Summary
The authors 3D-printed fiber-reinforced thermoplastic composites and investigated the evolution of impact toughness with several printing parameters (layer thickness, build orientation, fiber orientation, number of perimeters in the layering) and physical parameters (nature of the fiber, weight, volume, printing time, and cost). As highlighted by the authors, the impact toughness is barely reported in literature and additional data is desirable. However, the lengthy paper fails to contribute to the field. The impact of printing parameters has already been substantially published (at least in [27]), while the study of physical parameters lacks scientific soundness and are not well discussed. On the other hand, discussing cost and time parameters is an interesting contribution but should be better presented. The construction and writing of the article should be substantially improved.
Broad comments
- The authors should revise their manuscript with a native English speaker to clarify the article by removing redundancies and numerous irrelevant, unnecessary, or self-evident comments. Such improvements could facilitate the reading.
- The Introduction should be substantially modified to better introduce the subject and to provide enough background to the reader of a broader audience. A particular attention should be given to systematically defining the numerous abbreviations used by the authors (e.g. FRTP, FFF, CFF, etc.) and clearly and succinctly describing the concepts and methods related to the work (e.g. FFF printing, methods to print FRTPs using short or continuous fibers, etc.).
- The authors should improve Section 2 by more succinctly presenting the parameters and adding the description of materials tested (matrix, reinforcement types with short carbon fibers, long fibers, etc.). For example, the parameters studied could be listed in a Table instead of being literally described in a confusing way.
- As it is reported, it seems that only one sample was tested per serial number, which is insufficient to correctly assess the tendencies observed by the authors. At least three to five samples should be tested, and standard deviations should be given. Also, a more efficient labelling system could be used. Besides, results given in J could be converted in impact strength (J/m2) to be compared with literature (e.g. with results in Ref [27]).
- Overall, substantial improvements should be done for drawings, graphics, and tables to improve their readability. Many graphics are not necessary.
- Finally, authors attempt to correlate irrelevant variables and do not provide sufficient in-depth analyses. Briefly, relationships between weight or volume with absorbed energy are irrelevant, while relationships with time and cost are not enough discussed (see specific comments for detail).
Specific comments
- Line 40 - Please consider more specific references than [2-4] to illustrate the use of FRTP composites in industries. For example:
- For automotive: Ning, H., Janowski, G. M., Vaidya, U. K., & Husman, G. (2007). Thermoplastic sandwich structure design and manufacturing for the body panel of mass transit vehicle. Composite structures, 80(1), 82-91.
- For aerospace: http://dx.doi.org/10.1590/S1516-14392006000300002
- Line 47: Please revise the entire paragraph. The meaning is unclear and poorly introduces the subject of the article. A clear description of the existing additive manufacturing methods for FRTP composites would be more appropriate.
- Line 49: What does mean FFF and how references [12-14] are related to this terminology?
- Line 56: Why are Ref. [16-19] used for in this sentence? Ref 16 and 18 talk about residual stresses when machining alloys and steels, Ref 17 is a book describing the simulation of composites, and Ref 19 is a book on design of composites. How do these references illustrate that composites can have similar mechanical performances than metals (if this is the meaning of the authors)?
- Line 65: In which ways Ref. 29 and 30 are related to the present work?
- Line 83: It could be useful to explicit in this paragraph that high absorbed energies are desirable.
- Line 89: After mentioning that standardized metallic specimens used in Charpy test include a notch, the authors claimed that they did not include a notch for their specimen, without giving any justification. Why was this choice made? For example, 3D-printed FRTP composites were notched in [27] and notching samples is critical to control the crack propagation and assure repeatability.
- Line 103: the FFF and CFF technologies should have been described properly in the Introduction.
- Line 117: What percentage of solid infill (or fill density) was used in the composites?
- Table 1 and Table 2 could be combined to provide a better overview of the differences between the samples. Table 2 could be kept and extended with information provided in Table 1.
- Results and tendencies described from Line 141 to 150 seem to be very similar to those reported in Ref [27]. This should be mentioned by the authors.
- Figure 6 and 12 do not add any additional value to the reader. Please consider removing or amending them.
- Line 158: please clarify the sentence.
- For Figures 7 to 10. The way these Figures are currently presented is clearly misleading. First, captions should be changed: there is no relationship that can be established between weight, volume, printing time, or cost with absorbed energy. All these parameters are linked to the layering and should be correlated to this one only. One suggestion would be for the x-axis to include the serial number sorted by increasing energy and for the value of the energy to be placed next to the corresponding symbol on the graph. Else, rewording the caption is another option. More specifically:
- It seems that the authors based their weight and volume comparisons solely on data computed by the commercial software. While volume might be tricky to check with enough precision, weight could be easily verified experimentally. Was this verification done?
- The fact that the authors decided to present all their results as a function of absorbed energy is misleading. Weight and volume depend solely on the layering and it is obvious that the insignificant differences observed for weight and volume do not have any significant impact on the absorbed energy. The only relevant question about weight and volume is why a 0/90 deg. orientation lead to less material extruded than a 45/-45 deg. orientation. Do the authors have any suggestion?
- Similarly, absorbed energy depends solely on the layering in this configuration (and not on the weight or volume like Figure 7 and 8 tend to imply). A clear explanation of why a 45/-45 deg orientation has a higher impact resistance than a 0/90 deg orientation is currently missing and should be given.
- Finally, it is interesting to see if the configurations with the best impact resistance are economically interesting, like the authors attempted to do in Figure 9 and 10. But again, it should be absolutely clear for the reader that printing time/cost and absorbed energy are not correlated here (it is confusing as of now). Besides, a concluding remark highlighting this “ideal” configuration is missing in the text.
- In Section 4, Line 214, the description of the specimens is obscure. Do the author mean that only 8 layers out of 92 contain fibers in the first group, assuming that the 84 other layers are made of only matrix? If yes, the authors should clearly specify it and also describe their configuration within the composite.
- Figure 13 to 16 could be combined in one figure with different panels.
- Line 272: again, there is no direct link to be made between weight and absorbed energy here. If the authors want to corelate weight and absorbed energy, they should choose one layering configuration with varying only the infill density.
- Same comments than in Comment 12 can be done for Figures 17 to 20.
Author Response
Dear Reviewer,
Thank You for the review of our submitted manuscript. We have performed improvements of the manuscript based on your comments as following:
Brief Summary
The authors 3D-printed fiber-reinforced thermoplastic composites and investigated the evolution of impact toughness with several printing parameters (layer thickness, build orientation, fiber orientation, number of perimeters in the layering) and physical parameters (nature of the fiber, weight, volume, printing time, and cost). As highlighted by the authors, the impact toughness is barely reported in literature and additional data is desirable. However, the lengthy paper fails to contribute to the field. The impact of printing parameters has already been substantially published (at least in [27]), while the study of physical parameters lacks scientific soundness and are not well discussed. On the other hand, discussing cost and time parameters is an interesting contribution but should be better presented. The construction and writing of the article should be substantially improved.
Broad comments
- The authors should revise their manuscript with a native English speaker to clarify the article by removing redundancies and numerous irrelevant, unnecessary, or self-evident comments. Such improvements could facilitate the reading.
Response:
English in the article has been modified.
- The Introduction should be substantially modified to better introduce the subject and to provide enough background to the reader of a broader audience. A particular attention should be given to systematically defining the numerous abbreviations used by the authors (e.g. FRTP, FFF, CFF, etc.) and clearly and succinctly describing the concepts and methods related to the work (e.g. FFF printing, methods to print FRTPs using short or continuous fibers, etc.).
Response:
A description of the methods and an illustrative figure has been added to the Introduction.
- The authors should improve Section 2 by more succinctly presenting the parameters and adding the description of materials tested (matrix, reinforcement types with short carbon fibers, long fibers, etc.). For example, the parameters studied could be listed in a Table instead of being literally described in a confusing way.
Response:
Material properties were added through a new Table 1.
- As it is reported, it seems that only one sample was tested per serial number, which is insufficient to correctly assess the tendencies observed by the authors. At least three to five samples should be tested, and standard deviations should be given. Also, a more efficient labelling system could be used. Besides, results given in J could be converted in impact strength (J/m2) to be compared with literature (e.g. with results in Ref [27]).
Response:
We apologize that we forgot to state this information in the article. We added it to the text. Each series consisted of five specimens. Figures/graphs were adjusted and measured values were displayed for all samples. The absorbed energy [J] required to break the test specimen is evaluated according to the currently valid STN EN ISO 148-1 norm. The authors relied on this valid standard. Notch toughness as a proportion of the energy consumed to break the sample and the cross-sectional area [J/m2, J/cm2] has been used in the past.
- Overall, substantial improvements should be done for drawings, graphics, and tables to improve their readability. Many graphics are not necessary.
Response:
The figures and graphs in the article have been modified to improve their readability or have been removed.
- Finally, authors attempt to correlate irrelevant variables and do not provide sufficient in-depth analyses. Briefly, relationships between weight or volume with absorbed energy are irrelevant, while relationships with time and cost are not enough discussed (see specific comments for detail).
Response:
The monitored parameters, their relevance and their description in the text and figures were reworked and re-evaluated.
Specific comments
- Line 40 - Please consider more specific references than [2-4] to illustrate the use of FRTP composites in industries. For example:
- For automotive: Ning, H., Janowski, G. M., Vaidya, U. K., & Husman, G. (2007). Thermoplastic sandwich structure design and manufacturing for the body panel of mass transit vehicle. Composite structures, 80(1), 82-91.
- For aerospace: http://dx.doi.org/10.1590/S1516-14392006000300002
Response:
References have been added to the article.
- Line 47: Please revise the entire paragraph. The meaning is unclear and poorly introduces the subject of the article. A clear description of the existing additive manufacturing methods for FRTP composites would be more appropriate.
- Line 49: What does mean FFF and how references [12-14] are related to this terminology?
- Line 56: Why are Ref. [16-19] used for in this sentence? Ref 16 and 18 talk about residual stresses when machining alloys and steels, Ref 17 is a book describing the simulation of composites, and Ref 19 is a book on design of composites. How do these references illustrate that composites can have similar mechanical performances than metals (if this is the meaning of the authors)?
Response:
The entire paragraph was revised. A description of the methods has been added. References 16 and 18 have been removed. Additional relevant references have been added.
- Line 65: In which ways Ref. 29 and 30 are related to the present work?
Response:
References 29 and 30 were misplaced. They were moved to the right place.
- Line 83: It could be useful to explicit in this paragraph that high absorbed energies are desirable.
Response:
The information has been added to the article: “The absorbed energy K expresses the resistance of the material against the impact load, while the highest value is expected to achieve.”
- Line 89: After mentioning that standardized metallic specimens used in Charpy test include a notch, the authors claimed that they did not include a notch for their specimen, without giving any justification. Why was this choice made? For example, 3D-printed FRTP composites were notched in [27] and notching samples is critical to control the crack propagation and assure repeatability.
Response:
The authors consulted experts in the topic of bending impact tests. Both types of samples can be used for composites. The authors opted for non-notched specimens because deposition of long fiber reinforcement would be problematic at the notch. It would limit the investigated number of concentric rings.
- Line 103: the FFF and CFF technologies should have been described properly in the Introduction.
Response:
A description of the methods and an illustrative figure has been added to the Introduction.
- Line 117: What percentage of solid infill (or fill density) was used in the composites?
Response:
The 100% solid infill (or fill density) was used in the composites.
- Table 1 and Table 2 could be combined to provide a better overview of the differences between the samples. Table 2 could be kept and extended with information provided in Table 1.
Response:
The tables have been merged into one.
- Results and tendencies described from Line 141 to 150 seem to be very similar to those reported in Ref [27]. This should be mentioned by the authors.
Response:
A reference to the article has been added to the relevant paragraph.
- Figure 6 and 12 do not add any additional value to the reader. Please consider removing or amending them.
Response:
Relevant figures/graphs were adjusted and measured values were displayed for all samples.
- Line 158: please clarify the sentence.
Response:
The sentence has been modified.
- For Figures 7 to 10. The way these Figures are currently presented is clearly misleading. First, captions should be changed: there is no relationship that can be established between weight, volume, printing time, or cost with absorbed energy. All these parameters are linked to the layering and should be correlated to this one only. One suggestion would be for the x-axis to include the serial number sorted by increasing energy and for the value of the energy to be placed next to the corresponding symbol on the graph. Else, rewording the caption is another option. More specifically:
- It seems that the authors based their weight and volume comparisons solely on data computed by the commercial software. While volume might be tricky to check with enough precision, weight could be easily verified experimentally. Was this verification done?
Response:
The relevant captions have been changed. Information on the monitored parameters was added to the text of individual subchapters. Verification was performed by determining the weight of each sample. The differences between the values were negligible.
- The fact that the authors decided to present all their results as a function of absorbed energy is misleading. Weight and volume depend solely on the layering and it is obvious that the insignificant differences observed for weight and volume do not have any significant impact on the absorbed energy. The only relevant question about weight and volume is why a 0/90 deg. orientation lead to less material extruded than a 45/-45 deg. orientation. Do the authors have any suggestion?
Response:
An explanation has been added to subchapter 4.1.
- Similarly, absorbed energy depends solely on the layering in this configuration (and not on the weight or volume like Figure 7 and 8 tend to imply). A clear explanation of why a 45/-45 deg orientation has a higher impact resistance than a 0/90 deg orientation is currently missing and should be given.
Response:
The 45/-45 degree orientation has a higher impact resistance than the 0/90 degree orientation because the individual layers are better bonded with respect to the direction of the load.
- Finally, it is interesting to see if the configurations with the best impact resistance are economically interesting, like the authors attempted to do in Figure 9 and 10. But again, it should be absolutely clear for the reader that printing time/cost and absorbed energy are not correlated here (it is confusing as of now). Besides, a concluding remark highlighting this “ideal” configuration is missing in the text.
Response:
All confusing claims that could lead the readers to look for a correlation between production cost/printing time and absorbed energy have been modified. In case of specimens reinforced with chopped carbon, the best results were obtained with specimen type with parameters: 0.2 mm thickness, filament deposition 45/-45, base plane XY and two walls (briefly series no. 7 and 8).
In case of specimens reinforced with continuous fiber, authors recommend application of HSHT glass fiber deposited in concentric rings.
- In Section 4, Line 214, the description of the specimens is obscure. Do the author mean that only 8 layers out of 92 contain fibers in the first group, assuming that the 84 other layers are made of only matrix? If yes, the authors should clearly specify it and also describe their configuration within the composite.
Response:
The relevant paragraph in the text has been modified. A new Table 4 and Figure 10 have been added for better explanation.
- Figure 13 to 16 could be combined in one figure with different panels.
Response:
The original intention of the authors was the same, but smaller size of figures lead to the disappear of the fibre details in the breakage.
- Line 272: again, there is no direct link to be made between weight and absorbed energy here. If the authors want to corelate weight and absorbed energy, they should choose one layering configuration with varying only the infill density.
Response:
Authors decided to usage the same layering configuration for each series of specimens.
- Same comments than in Comment 12 can be done for Figures 17 to 20.
Response:
The relevant captions have been changed. Information on the monitored parameters was added to the text of individual subchapters.
Reviewer 2 Report
The paper is very interesting and the topic of the paper falls within the scope of the journal. Overall, the paper is well organized and written. The authors are encouraged to deal with the following comments before the manuscript could be accepted:
- If possible, please refer the following paper (doi.org/10.1016/j.compositesb.2019.107191) to analyze the failure progress.
- It it more appreciate that the authors can establish the finite element modeling corresponding to the experimental test.
Author Response
Dear Reviewer,
Thank You for the review of our submitted manuscript. We have performed improvements of the manuscript based on your comments as following:
Comments and Suggestions for Authors
The paper is very interesting and the topic of the paper falls within the scope of the journal. Overall, the paper is well organized and written. The authors are encouraged to deal with the following comments before the manuscript could be accepted:
1.
If possible, please refer the following paper (doi.org/10.1016/j.compositesb.2019.107191) to analyze the failure progress.
Response:
The reference to the requested article has been read and added to the article. We used the following paper in analysis of failure progress.
2.
It is more appreciate that the authors can establish the finite element modeling corresponding to the experimental test.
Response:
We originally planned to perform the Finite Element analysis of the experiment. Due to time limitations and COVID disease, we did not conduct the analysis. However, we would like to implement it in the future.
Reviewer 3 Report
Overall, the findings are consistent, but there are a few shortcomings from my point of view: The first part describing the tests with short fiber reinforced specimen there is no comparison with a not-reinforced specimen (just the matrix material). The results however are clearly presented and comprehensible. The second part concerning the continuous fibers is very confusing, as many parameters are varied. The diagrams are quite complex and need a thorough understanding to follow along with the findings stated in the paper. Also, Figures 17, 18, 19 and 20 are missing a data point for the specimen “[1] K UNI 8 direct”. In general, the experiments could be described better to make it more understandable for the reader. The introduction contains a description of the experimental setup for the impact test. That should be in a different /own section. Also, there is no description on how many specimen per parameter setting were tested. It is unclear if there is any statistical relevance behind the numbers given in the whole paper! The group decided to test a lot of different fiber types, printing orientations, but only two fiber loadings. From my point of view, it would make more sense to try different volume fractions and less fiber types and report those individually. That would also make the paper easier to follow through for the reader. In the Conclusion it is stated that: “However, the suitable selection of the lamina thickness depends on the load direction. It means the properties of this material are independent of the direction.“ From my point of view this should be “dependent”. The English language could also be checked by a native speaker, if possible. Especially at the beginning of the paper there are a few problems with grammar.
Author Response
Dear Reviewer,
Thank You for the review of our submitted manuscript. We have performed improvements of the manuscript based on your comments as following:
Comments and Suggestions for Authors
Overall, the findings are consistent, but there are a few shortcomings from my point of view: The first part describing the tests with short fiber reinforced specimen there is no comparison with a not-reinforced specimen (just the matrix material). The results however are clearly presented and comprehensible. The second part concerning the continuous fibers is very confusing, as many parameters are varied. The diagrams are quite complex and need a thorough understanding to follow along with the findings stated in the paper. Also, Figures 17, 18, 19 and 20 are missing a data point for the specimen “[1] K UNI 8 direct”. In general, the experiments could be described better to make it more understandable for the reader. The introduction contains a description of the experimental setup for the impact test. That should be in a different /own section. Also, there is no description on how many specimen per parameter setting were tested. It is unclear if there is any statistical relevance behind the numbers given in the whole paper! The group decided to test a lot of different fiber types, printing orientations, but only two fiber loadings. From my point of view, it would make more sense to try different volume fractions and less fiber types and report those individually. That would also make the paper easier to follow through for the reader. In the Conclusion it is stated that: “However, the suitable selection of the lamina thickness depends on the load direction. It means the properties of this material are independent of the direction. “ From my point of view this should be “dependent”. The English language could also be checked by a native speaker, if possible. Especially at the beginning of the paper there are a few problems with grammar.
In detail:
1.
The first part describing the tests with short fiber reinforced specimen there is no comparison with a not-reinforced specimen (just the matrix material).
Response:
We did not perform experimental testing on unreinforced specimens, because we don’t have the necessary material in stock. The purchase of nylon would take a few months. These tests were performed by the authors of other articles, and we became acquainted with the issue.
2.
The diagrams are quite complex and need a thorough understanding to follow along with the findings stated in the paper.
Response:
Changes have been made to the article. We have tried to simplify this part of the article and make it more clearly for readers.
3.
Figures 17, 18, 19 and 20 are missing a data point for the specimen “[1] K UNI 8 direct”.
Response:
The mentioned specimen type overlaps with specimen “[2] K UNI 8 perpend” in the figure.
4.
The introduction contains a description of the experimental setup for the impact test. That should be in a different /own section.
Response:
The request was accepted. The description of the experimental setup has its section in the paper.
5.
Also, there is no description on how many specimen per parameter setting were tested. It is unclear if there is any statistical relevance behind the numbers given in the whole paper!
Response:
We apologize that we forgot to state this information in the article. We added it to the text. Each series consisted of five specimens.
6.
The group decided to test a lot of different fiber types, printing orientations, but only two fiber loadings. From my point of view, it would make more sense to try different volume fractions and less fiber types and report those individually.
Response:
The aim of the authors was to bring complete information about the test results with regard to various printing parameters.
7.
In the Conclusion it is stated that: “However, the suitable selection of the lamina thickness depends on the load direction. It means the properties of this material are independent of the direction. “ From my point of view this should be “dependent”.
Response:
We apologize for the translation error. The statement was adjusted to correct form: “However, the suitable selection of the lamina thickness depends on the load direction. It means the properties of this material are dependent of the direction.“
Round 2
Reviewer 1 Report
Brief summary
The authors provided a substantially modified version of their manuscript with noticeable improvements in the Introduction, the English quality, and in clarifying the authors’ analyses. However, many concerns raised in the first round have not been fully addressed. The article would significantly gain in readability and impact if the content would be better presented in a much more succinct manner.
Response to Broad comments
- English has been substantially modified and the article is now more readable. However, the authors should continue the effort or seek help from editing companies as many sentences are still confusing and many vocabulary and punctuation mistakes are still there.
- The Introduction is now better describing the subject but could certainly be further improved (for example by detailing the similar studies cited by the authors (ref. 30 to 33) and by explaining how the authors provide additional information to these studies).
- Table 2 provides a clearer presentation of printing parameters studied in authors’ work. However, there is still confusing information about Table 1. I understand from the modifications that only one matrix is studied: nylon. Then, mechanical properties of the nylon could be added to Table 1. The other fibers mentioned in the Table are short or long fibers? Were all fibers used by CFF and FFF? This should be mentioned in the Table to help the reader to follow.
- If not mistaken, the number of samples tested for each series is still not mentioned in the article (it should at least be mentioned in the caption of Table 3 and Figures and in the paragraph introducing the results (around line 170). Standard deviations should be given in Table 3. I do not understand the authors’ answer: “Notch toughness as a proportion of the energy consumed to break the sample and the cross-sectional area [J/m2, J/cm2] has been used in the past.” Does that mean that the authors do not want to use this parameter because it has been used in the past?
- Figures were modified to improve readability but could still be significantly improved.
- Authors substantially re-evaluated the impact of parameters.
Response to Specific comments:
- Please revise the grammar of the added sentences in Line 110, as it is unclear in the present form.
- The authors should clearly justify/assume in the article (Line 117) their choice for not using a notch. I still have concerns with not notching the samples. Many studies on FRTP used notched samples and one advantage of 3D-printing, as claimed by the authors, is to manufacture samples with more complex geometries. Short printing paths are clearly visible in Figure 11 – bottom image. Thus, I don’t see how printing samples with a notch is difficult (there is no need for changing the number of walls).
- This information should be given in the article. That was the purpose of question 7.
- Adding the values for all samples is an interesting addition. However, it is necessary to add standard deviations calculated from the samples into Table 3 and Table 5. Despite this addition to Figure 7 and 12, the relevance of these Figures is still weak, as it is just a graphical representation of Tables 3 and 5, respectively. The results are already graphically represented in Figure 8 and 9 (for Figure 7) and in Figure 17, 18, and 19 (for Figure 12) in a more meaningful way.
- The new captions are much clearer and less misleading. The additional succinct information in each subchapter is appreciated. However, please consider using “filament orientation” instead of “filament deposition” to describe the layering direction.
-
- I believe that the sentences in Line 169 were added in response to question 12. Please revise the grammar and the sentence to better explain that the values computed by the commercial software (i.e. weight, printing time) were experimentally verified and negligible differences were observed.
- Please revise the grammar of the explanation added in Line 178. As written, the reader could believe that the orientation has an impact on the adhesion between layers. The 45/-45 degree orientation has a higher resistance because all layers counter the crack propagation, while in a 0/90 degree orientation, only 50% of layers efficiently counter the crack propagation while the other 50% of layers facilitate the crack propagation.
- Please revise the grammar of the explanation added in Line 209. What does “storage path” means and what is the “set G-code from manufacturer”. Do the authors mean that the G-code generated for the 45/-45 degree orientation is better filling the printing area compared to the 0/90 degree orientation, thus leading to more material extruded, longer printing time, and more weight? If that is the case, that would be interesting to provide relevant data to support this assertion.
- Table 4 is much clearer and is highly appreciated. Please detail in the caption of Figure 10 what are (a) and (b). Please revise the description of samples in the text as well. For example (or something similar):
“Two configurations were studied: (i) a ‘sandwiched’ configuration with 8 reinforced layers equally distributed in the vertical direction (on a total of 100 layers); (ii) a ‘plain’ configuration with only reinforced layers (92 layers in total). Both configurations have a top and bottom part made of 4 layers of nylon matrix.”
-
- Line 244: “The selected matrix material was Onyx”. If not mistaken, Onyx here refers to the brand of the material. If nylon is the material, then it should be mentioned: nylon (from Onyx) or something similar. Same observation when the authors use Onyx fibers (e.g. Table 1). It should be mentioned: chopped carbon fiber (Onyx).
- Please use (a), (b), and (c) sub-captions for Figure 11. Are the infill patterns presented in Figure 11 identical for each layer? 3D-printer software usually altern the layer directions with a 90 degree angle to improve mechanical resistance (at least for the top and bottom images). If the layers are identical, please use other terms than unidirectional and isotropic terminologies, as they are not relevant. Longitudinal, concentric, and 45-degree orientations could be more appropriate (or something similar).
- Using (a), (b), and (c) panels does not require to reduce the size of the images (cf. Figure 11).
The other questions have been addressed in a more or less satisfying way.
Author Response
Dear Reviewer,
Thank You for the review of our submitted manuscript. Individual reviewers had various comments on some parts of the article (even contradictory). We have tried to make changes and improvements in the article based on the compromise and your comments.
Comments and Suggestions for Authors
Brief summary
The authors provided a substantially modified version of their manuscript with noticeable improvements in the Introduction, the English quality, and in clarifying the authors’ analyses. However, many concerns raised in the first round have not been fully addressed. The article would significantly gain in readability and impact if the content would be better presented in a much more succinct manner.
Response to Broad comments
- English has been substantially modified and the article is now more readable. However, the authors should continue the effort or seek help from editing companies as many sentences are still confusing and many vocabulary and punctuation mistakes are still there.
Response:
English in the article has been modified.
- The Introduction is now better describing the subject but could certainly be further improved (for example by detailing the similar studies cited by the authors (ref. 30 to 33) and by explaining how the authors provide additional information to these studies).
Response:
The authors described similar studies in more detail and stated the contributions of the article.
- Table 2 provides a clearer presentation of printing parameters studied in authors’ work. However, there is still confusing information about Table 1. I understand from the modifications that only one matrix is studied: nylon. Then, mechanical properties of the nylon could be added to Table 1. The other fibers mentioned in the Table are short or long fibers? Were all fibers used by CFF and FFF? This should be mentioned in the Table to help the reader to follow.
Response:
Pure nylon was not included in this study, so we did not specify its mechanical properties. Instead of nylon, we used Onyx, which is defined by the manufacturer as nylon filled/reinforced with chopped carbon fibre. In the case of Chapter 3, the specimens were made only from Onyx by the FFF method. In Chapter 4, the specimens were reinforced with long fibres (such as Carbon, Kevlar, Glass and HSHT Glass). The matrix material was Onyx. Printing of long-fibre reinforced specimens is solely possible by the CFF method. To make the paper clearer to readers, we've modified the text and provided additional information in Table 1.
- If not mistaken, the number of samples tested for each series is still not mentioned in the article (it should at least be mentioned in the caption of Table 3 and Figures and in the paragraph introducing the results (around line 170). Standard deviations should be given in Table 3. I do not understand the authors’ answer: “Notch toughness as a proportion of the energy consumed to break the sample and the cross-sectional area [J/m2, J/cm2] has been used in the past.” Does that mean that the authors do not want to use this parameter because it has been used in the past?
Response:
Each test series for measurement consisted of five specimens. This is stated at the beginning of the third chapter. The standard deviation was added to Tables 3 and 5. Also, the absorbed energy is given in [J] and [J/m2] units.
- Figures were modified to improve readability but could still be significantly improved.
Response:
We have made additional modifications to increase the readability of the figures.
- Authors substantially re-evaluated the impact of parameters.
Response:
The authors aim is to provide a sufficient in-depth analysis of the influence of various parameters on the impact toughness of the specimen. We implement any comments that lead to an increase in article quality.
Response to Specific comments:
- Please revise the grammar of the added sentences in Line 110, as it is unclear in the present form.
Response:
The sentence in Line 110 (previous version) has been modified.
- The authors should clearly justify/assume in the article (Line 117) their choice for not using a notch. I still have concerns with not notching the samples. Many studies on FRTP used notched samples and one advantage of 3D-printing, as claimed by the authors, is to manufacture samples with more complex geometries. Short printing paths are clearly visible in Figure 11 – bottom image. Thus, I don’t see how printing samples with a notch is difficult (there is no need for changing the number of walls).
Response:
Because previous studies [35,36] were performed on notched specimens, the authors considered a similar type of specimen. The authors decided to perform the experiment on non-notched samples after careful consideration. The reason is the significant influence of the notch on the mechanical behaviour of the specimen under impact loading and the problematic deposition of reinforcement in the notch area. The fibre rings would copy the shape of the notch in the case of a notched sample. The strength of the material in the notch area will be reduced due to the change in the direction of the fibre and the fibres would be loaded by bending stress. The deposition of the fibre around the notch will show inhomogeneity. Images from the slicing software also show the problem around the notch.
- a)
- b)
- c)
- The printer starts printing at the notch location in the case of notched samples. As printer users we cannot influence this. According to the authors, this is an inappropriate fact that will negatively affect the energy absorption of the sample during the test. The authors therefore decided to use a specimen without a notch, where the beginning and end of the filament are stored outside the impact force.
- According to the authors, the placement of the notch will negatively affect the efficiency of fibre deposition around the notch. The fibre changes direction, it probably breaks and thus its strength is negatively affected. The end of the fibre is located around the notch in the case of a longitudinally arranged fibre.
- As with the longitudinally arranged fibre, the notch appears to be a problematic element of the specimen in the case of concentric rings. Two of the four rings are located in a completely inappropriate way. The solution would be to use only two concentric rings. In such a case, however, the equivalence of the volume fraction of fibre in the lamina would not be maintained. In the case of longitudinal or 45/-45 degree fibre orientation the user does not have the opportunity to influence the proportion of fibre in the lamina. This option is only in the case of concentric rings, when it is up to the user how many circuits he chooses.
- This information should be given in the article. That was the purpose of question 7.
Response:
The fill density parameter is specified in Table 2.
- Adding the values for all samples is an interesting addition. However, it is necessary to add standard deviations calculated from the samples into Table 3 and Table 5. Despite this addition to Figure 7 and 12, the relevance of these Figures is still weak, as it is just a graphical representation of Tables 3 and 5, respectively. The results are already graphically represented in Figure 8 and 9 (for Figure 7) and in Figure 17, 18, and 19 (for Figure 12) in a more meaningful way.
Response:
The standard deviations were added to Tables 3 and 5. Figures 7 and 12 (previous version) have been removed.
- The new captions are much clearer and less misleading. The additional succinct information in each subchapter is appreciated. However, please consider using “filament orientation” instead of “filament deposition” to describe the layering direction.
Response:
The term “filament orientation” has been used in the article.
- I believe that the sentences in Line 169 were added in response to question 12. Please revise the grammar and the sentence to better explain that the values computed by the commercial software (i.e. weight, printing time) were experimentally verified and negligible differences were observed.
Response:
The sentences in the text have been modified.
- Please revise the grammar of the explanation added in Line 178. As written, the reader could believe that the orientation has an impact on the adhesion between layers. The 45/-45 degree orientation has a higher resistance because all layers counter the crack propagation, while in a 0/90 degree orientation, only 50% of layers efficiently counter the crack propagation while the other 50% of layers facilitate the crack propagation.
Response:
The sentences in Line 178 have been modified.
- Please revise the grammar of the explanation added in Line 209. What does “storage path” means and what is the “set G-code from manufacturer”. Do the authors mean that the G-code generated for the 45/-45 degree orientation is better filling the printing area compared to the 0/90 degree orientation, thus leading to more material extruded, longer printing time, and more weight? If that is the case, that would be interesting to provide relevant data to support this assertion.
Response:
Since G-code is not available, the authors performed an analysis of images from the slicing software.
45/-45 filament orientation:
1 px = 0.031218 mm
|
Type Type of line |
Number of pixels |
Line length [mm] |
Number of lines |
Lines length [mm] |
|
Horizontal |
19 |
0.593139 |
92 |
54.56878 |
|
Vertical |
19 |
0.593139 |
14 |
8.36478 |
|
Diagonal |
264 |
11.74064 |
94 |
1103.62 |
|
Horizontal short |
11 |
0.343396 |
2 |
0.686792 |
|
Vertical short |
5 |
0.222361 |
2 |
0.444721 |
Total length of lines in one lamina is: 1167.685 mm.
If laminate cosists of one houndred laminas the total length of lines is 116 768.5 mm.
0/90 filament orientation:
1 px = 0.049639 mm
|
Type of line |
Number of pixels |
Line length [mm] |
Number of lines |
Lines length [mm] |
|
Horizontal |
7 |
0.426829 |
132 |
56.34146 |
|
Vertical |
136 |
8.292683 |
133 |
1102.927 |
Total length of lines in one lamina is: 1159.268 mm.
1 px = 0.060976 mm
|
Type of line |
Number of pixels |
Line length [mm] |
Number of lines |
Lines length [mm] |
|
Horizontal |
8 |
0.397112 |
20 |
7.942238 |
|
Vertical |
1080 |
53.61011 |
21 |
1125.812 |
Total length of lines in one lamina is: 1133.755 mm.
Then the total length of lines is 50*1159.268 + 50*1133.755 = 114 651.15 mm.
- Table 4 is much clearer and is highly appreciated. Please detail in the caption of Figure 10 what are (a) and (b). Please revise the description of samples in the text as well. For example (or something similar): “Two configurations were studied: (i) a ‘sandwiched’ configuration with 8 reinforced layers equally distributed in the vertical direction (on a total of 100 layers); (ii) a ‘plain’ configuration with only reinforced layers (92 layers in total). Both configurations have a top and bottom part made of 4 layers of nylon matrix.”
Response:
The authors added a more accurate caption of Figure 10 (previous version). In addition, a description of the specimens was added to the text of the article.
- Line 244: “The selected matrix material was Onyx”. If not mistaken, Onyx here refers to the brand of the material. If nylon is the material, then it should be mentioned: nylon (from Onyx) or something similar. Same observation when the authors use Onyx fibers (e.g. Table 1). It should be mentioned: chopped carbon fiber (Onyx).
Response:
Onyx is a trademark of a material that consists of nylon reinforced with chopped carbon fibre in fibrous form. The authors after consistent consideration used following term: nylon reinforced with chopped fibre. Also, we have described the material in the article.
- Please use (a), (b), and (c) sub-captions for Figure 11. Are the infill patterns presented in Figure 11 identical for each layer? 3D-printer software usually altern the layer directions with a 90 degree angle to improve mechanical resistance (at least for the top and bottom images). If the layers are identical, please use other terms than unidirectional and isotropic terminologies, as they are not relevant. Longitudinal, concentric, and 45-degree orientations could be more appropriate (or something similar).
Response:
We apologize for forgetting to specify the sub captions in Figure 11. In the case of continuous fibre deposition strategy, the slicing software allows the user to modify the fibre orientation for each lamina. The manufacturer states just mentioned names of the fibre deposition strategies. After careful consideration, the authors decided to comply with the comment and applied the terms: longitudinal, concentric and 45-degree orientation.
- Using (a), (b), and (c) panels does not require to reduce the size of the images (cf. Figure 11).
The other questions have been addressed in a more or less satisfying way.
Response:
We apologize; we did not initially understand the comment. We accepted the reviewer comment and the given pictures are combined into one figure with panels a, b and c (and d).
Reviewer 2 Report
The paper can be accepted at current form.
Author Response
Dear Reviewer,
Thank You for the review of our submitted manuscript. Individual reviewers had various comments on some parts of the article (even contradictory). We have tried to make changes and improvements in the article based on the compromise and your comments.
Comments and Suggestions for Authors
The paper can be accepted at current form.
Reviewer 3 Report
The authors have addressed all reviewers' comments and significantly improved the manuscript.
Author Response
Dear Reviewer,
Thank You for the review of our submitted manuscript. Individual reviewers had various comments on some parts of the article (even contradictory). We have tried to make changes and improvements in the article based on the compromise and your comments.
Comments and Suggestions for Authors
The authors have addressed all reviewers' comments and significantly improved the manuscript.